# Validated Methods for Inactivation of Tick-Borne Encephalitis Virus Compatible with Immune-Based and Enzymatic Downstream Analyses

**DOI:** 10.3390/v17060810

**Published:** 2025-06-03

**Authors:** Simone Leoni, Stephen L. Leib, Katharina Summermatter, Denis Grandgirard

**Affiliations:** 1Institute for Infectious Diseases, University of Bern, 3001 Bern, Switzerland; simone.leoni@unibe.ch (S.L.); stephen.leib@unibe.ch (S.L.L.); katharina.summermatter@unibe.ch (K.S.); 2Multidisciplinary Center for Infectious Diseases, University of Bern, 3001 Bern, Switzerland; 3Graduate School for Cellular and Biomedical Sciences, University of Bern, 3001 Bern, Switzerland

**Keywords:** Tick-Borne Encephalitis Virus (TBEV), flavivirus, inactivation, immuno-assay, enzymatic assays, biosafety

## Abstract

Tick-Borne Encephalitis Virus (TBEV) is impacting public health in the Eurasian region, with increasing case numbers. There is, therefore, a need to expand research efforts and the corresponding infrastructure capacity. Since TBEV is classified as a risk group 3 organism in Switzerland, handling infectious material containing the virus is restricted to biosafety level 3 laboratories. In some instances, downstream analyses may need to be performed outside of the containment facility. It is, therefore, essential to validate effective inactivation protocols compatible with the safe and accurate processing of samples. This study evaluated UV irradiation, chemical treatment with detergents, and mechanical filtration as candidate methods to inactivate TBEV infectious samples, including culture supernatants and tissue homogenates, while preserving their compatibility for different assays. Among the methods tested, 45 s of UV irradiation or Triton-X100 at concentrations between 0.05% and 0.1% effectively inactivated TBEV while mostly preserving the integrity of the processed samples for immuno- or enzymatic assays. These findings establish safe and reliable procedures for advancing TBEV research beyond high-containment settings.

## 1. Introduction

Arthropod-borne flaviviruses, including Zika virus (ZIKV), West Nile virus (WNV), Dengue virus, Yellow Fever virus (YFV), and Tick-Borne Encephalitis virus (TBEV), pose an increasing threat to global health, with millions of cases reported annually. TBEV is currently the most medically relevant tick-borne flavivirus and is a growing public health challenge in the Eurasian region, where it is endemic and accounts for more than 10,000 cases each year [1].

TBEV is classified as a risk group 3 pathogen in several countries, including Switzerland [2], due to its potential to cause severe neurological disease with long-term complications or death, and the limited efficacy of available vaccines as post-exposure prophylaxis. The virus is highly infectious, with a documented risk of aerosol transmission in laboratory settings and reported cases of lab-acquired infections [3,4]. Furthermore, its stability in biological fluids heightens the risk of accidental contamination, reinforcing the need for strict biosafety measures [5,6]. Therefore, it must be handled in containment laboratories of biosafety level 3 (BSL-3) with strict biosafety protocols to minimize the risk of exposure and ensure safe processing. Downstream analyses such as genomics and proteomics require sophisticated or cost-intensive equipment that is often a part of core facilities outside of the BSL-3 environment, making the inactivation of samples a prerequisite before analysis can be performed.

Protocols for the isolation of DNA and RNA from infectious samples are well established and usually show the efficient inactivation of enveloped viral particles during the process [7,8,9]. One of the most widely used approaches is based on the guanidine thiocyanate–phenol–chloroform protocol, which extracts RNA, DNA, and proteins in different phases [10]. However, this method has some drawbacks, including the toxicity of the reagents, lengthy protocols, degradation, and poor protein solubilization [11]. Additionally, it remains unclear whether the protein yield and quality achieved with this method match those obtained using specialized protein-focused extraction procedures [12]. Consequently, there is a need to develop and validate other rapid, non-toxic viral inactivation protocols specifically optimized for downstream protein quantification and enzymatic analyses. In this study, our primary goal was to develop efficient TBEV inactivation methods that would allow accurate protein-based assays, such as immunoassays and enzymatic assays, to be performed safely outside BSL-3 laboratories. These assays are commonly used in research settings handling cell cultures, tissues, or in vivo models to evaluate various aspects of the host response to the virus.

Previous studies have explored various reagents and physical conditions for TBEV inactivation. Complete inactivation was achieved after 20 min at pH 2 or 3. Heat inactivated the virus after 5 min at 55 °C but not at 72 °C for 15 s. In this case, the efficiency of inactivation depended on the sample composition. UV inactivation was dependent on the probe volume and composition and was usually achieved in a range between 10 and 30 min. Alcohol inactivation was effective with 40% ethanol or methanol and 30% 2-propanol after 10 min. Chemical inactivation was also successful with 20 min of incubation using 0.2% Triton X-100 or sodium deoxycholate, 5% Tween, or 1% paraformaldehyde [13,14,15]. Several other flaviviruses and enveloped viruses like Zika virus (ZIKV) [16], Usutu virus [17], Hepatitis C virus [18], West Nile virus (WNV) [19], and SARS-CoV-2 [8] can also be inactivated under similar conditions, although the exposure times and concentrations may vary. Despite the successful validation of the inactivation of these viruses, their compatibility with downstream analyses is seldom tested in these studies.

After the establishment of efficient inactivation methods for TBEV, we also evaluated the suitability of the inactivated samples for downstream analyses, with a focus on immune-based and enzymatic methods, exemplified by Luminex measurements and a lactate dehydrogenase (LDH) assay. Different types of samples were tested, including cell culture supernatants, cell extracts, or tissue homogenates containing different protein concentrations. This work proposes effective inactivation protocols preserving the integrity of the samples for further research with TBEV and potentially other flaviviruses.

## 2. Materials and Methods

### 2.1. Biosafety Statement

All experiments involving the handling of infectious TBEV-containing samples were performed under BSL-3 conditions with strict biosafety protocols to minimize potential risks. All procedures were carried out in a certified biological safety cabinet. All tissue culture flasks or plates containing infectious material were enclosed in Polysteriboxes (Ritter Medical, Schwabmünchen, Germany) for incubation, and all infectious wastes were autoclaved before disposal.

Permission to work with TBEV was granted by the Swiss Federal Office of Public Health (A192665/3) with the consultation of the Federal Office for the Environment, the Federal Food Safety and Veterinary Office, and the Swiss Expert Committee for Biosafety.

### 2.2. Animal Ethics

Brain tissues used as tested samples were collected from ongoing in vivo experiments on TBEV-infected and control infant rats. These experiments were approved by the Animal Care and Experimentation Committee of the Canton of Bern, Switzerland (license no. BE33/2022).

### 2.3. Viral Stock and Preparation of Infectious Samples of Different Types

The infection experiments were performed with the European subtype of TBEV Hypr. The amplification was conducted on A549 cells grown in minimum essential medium (Thermofisher, Waltham, MA, USA, #11095) supplemented with 10% fetal bovine serum (Sigma Aldrich, St. Louis, MO, USA, #F9665), 1.25% GlutaMAX (Thermofisher, #35050), 1% Non-Essential Amino Acids (Thermofisher, #11140), and 1% Antibiotic–Antimycotic (Thermofisher, #15240). The cells were incubated at 37 °C and 5% CO_2_. At 3 days post-infection, the infected cells were scrapped from the surface of the flask and collected together with the supernatant in a tube ready for snap-freezing at −80 °C in order to lyse the cells. Tissue homogenates were prepared from the brains of control or infected infant rats. The isolated tissues were supplemented with 6-times-volume (*w*/*v*) saline solution before being homogenized at 4500 rpm for 30 s (2×) in screw-cap tubes with an O-ring in the lid, containing 1.4 mm zirconium beads, using a bead homogenizer (Precellys Evolution, Bertin Technologies, Montigny-le-Bretonneux, France). After centrifugation (13,000 rpm for 10 min seconds at 4 °C) in a microcentrifuge Biofuge Fresco (Heraeus, Hanau, Germany), cleared supernatant was collected in fresh tubes and stored at −80 until further use.

### 2.4. BCA Assay

Protein concentrations were determined using the bicinchoninic acid (BCA) method for both cell media (at dilutions of 1:1, 1:10, 1:50, and 1:100) and brain homogenates (at dilutions of 1:5, 1:50, 1:250, and 1:500) using the Pierce BCA protein assay kit (Thermofisher, #A55864) according to manufacturer instructions. Absorbance was measured at 562 nm using a BioTek Cytation 5 plate reader (Agilent Technologies, Santa Clara, CA, USA).

### 2.5. Viral Inactivation via UVC Light

In total, 100 µL of cell culture supernatant or brain homogenate (either kept undiluted or diluted 1:5 with saline) was pipetted in a 1.5 mL microtube EasyFit PP Clear (TreffLab, Degersheim, Switzerland, #96.07246.9.01) and put horizontally on a 5 cm high (about 1.97 in) ice-containing tray. Similar inactivation procedures with small volumes of different viral suspensions (Mpox and SARS-CoV2) placed inside microtubes have been shown to be efficient, supporting the hypothesis that enough UVC radiation can penetrate the plastic material [20,21]. Thereafter, the tray with the samples was introduced into the chamber of a UVC crosslinker, equipped with 6 × 8 W bulbs that irradiate at 254 nm (Analytik Jena, Jena, Germany, CL-3000). The distance between the samples and the light sources was approximately 15 cm. Irradiation was performed for different durations between 15 s (~100 mJ/cm^2^ total energy exposure) and 10 min (~4400 mJ/cm^2^). The inactivated samples were then tested for their infectivity via plaque assay. The instrument measures the effective irradiation amount using internal sensors.

### 2.6. Viral Inactivation via Chemical Reagents

Cell culture supernatants or brain homogenates were treated with Triton X-100 (TX-100) or Tween-20 at different concentrations. For TX-100, we tested concentrations between 0.025 and 0.2%, while for Tween-20, we tested concentrations between 0.05 and 10%. All concentrations are expressed as the final concentration in the sample. The samples were incubated for 20 min at room temperature before their infectivity was assessed via plaque assay.

### 2.7. Viral Depletion via Column Filtration

This method was applied only to culture supernatants since homogenates would rapidly clog the filter. In total, 100 µL of culture supernatant was filtered through Sartorious Vivaspin columns with different pore sizes (100 kDa (VS0142) or 300 kDa (VS0152)), according to manufacturer instructions. After filtration, the remaining unfiltered volume and the flow-through were frozen at −80 °C for later testing of the infectivity via TCID_50_ or plaque assay.

### 2.8. Plaque Assay

Plaque assays were performed on A549 cells. Briefly, 6-well plates were seeded with A549 cells at a density of 3.3 × 10^5^ cells/well and incubated for two days to reach confluence at 37 °C and 5% CO_2_. The culture medium was replaced with 1 mL maintenance medium (MEM, 2% FBS, 1.25% GlutaMAX, 1% NEAA, and 1% Antibiotic–Antimycotic) in each well. In total, 100 µL of UV-inactivated probes, or 10 µL of TX-100 and Tween-20-inactivated probes, were diluted in 1 mL maintenance medium and added to the well (1 mL per well), obtaining a final volume of 2 mL. A higher dilution was necessary with TX-100-inactivated samples to avoid cytotoxicity due to the inactivating agent. The plates were incubated for 1.5 h at 37 °C and 5% CO_2_. Thereafter, the inoculum was removed, 2 mL of 1:1 Avicel (RC-581NF) DMEM (2×) supplemented with 2% FBS was added, and the plates were incubated for 5 days. The medium was removed, and the cells were washed once with 2 mL of PBS and fixed for 20 min with 4% PFA in PBS (Roti^®^Histofix, Roth AG, Arlesheim, Switzerland). Cell monolayers were stained with 1 mL of 1% Crystal violet in dH_2_O/well for 20 min at room temperature. Thereafter, the plates were washed 3 times with tap water and imaged (Vilber, Fusion FX, Eberhardzell, Germany). The presence and/or the number of plaques was finally determined.

### 2.9. TCID_50_ Assay

Viral titer was assessed by TCID_50_ on Vero cells. Then, 96-well plates were seeded with ≈ 20,000 cells at 100 μL/well and incubated overnight at 37 °C and 5% CO_2_ to reach confluence. Thereafter, the medium was removed and replaced by 100 μL of serial decimal dilutions of the sample of up to 10^−8^. The plates were incubated for 3 days at 37 °C and 5% CO_2_. The medium was removed, and the whole plate was submerged in 4% PFA in PBS for 20 min for virus inactivation and fixation. The plates were then washed two times with PBS before incubation with 0.1% Saponin in PBS for 30 min at 4 °C. Mouse monoclonal pan-flavivirus anti-envelope glycoprotein 4G2 antibody prepared from hybridoma cells (ATCC HB-112, provided by Prof. M. Alves, Institute of Virology, University of Bern) was used as the primary antibody for viral detection. The wells were incubated with a 1:10 dilution of hybridoma supernatant in PBS containing 0.3% Saponin for 45 min at 37 °C. After two washings with 0.1% Saponin, wells were incubated with a 1:250 dilution of secondary antibody (Rabbit anti-mouse HRP; DAKO #P0260) in PBS containing 0.3% Saponin for 45 min at 37 °C, followed by two washings with 0.1% Saponin in PBS. The colorimetric enzymatic reaction was performed by adding 50 µL of 3-amino-9-ethylcarbazole (AEC) substrate/well for 30-45 min. In total, 50 mL of AEC substrate was composed of 47.5 mL of 50 mM acetate buffer (pH = 5.0), 2.5 mL of AEC solution (1 tablet of 3-amino-9-ethylcarbazol (Sigma-Aldrich; A5754) in 2.5 mL of N-,N-Dimethylformamide) and 25 ul of hydrogen peroxide (30% H_2_O_2_) (Carl Roth, Karlsruhe, Germany, 8070.2). The plates were finally washed with tap water, and the titer was determined according to the Reed and Münch calculation method [22].

### 2.10. Luminex Assay

In total, 160 µL of uninfected cerebellar homogenate, prepared as described above from an uninfected animal, was supplemented with 640 µL of saline solution to achieve a 1:5 dilution. The diluted homogenate, as well as the cell culture supernatant, was spiked with a standard solution provided by the Luminex kit, containing five cytokines (IFN-γ, IL-6, TNF-α, IL-1β, and IL-10) of known concentrations. One-hundred-microliter aliquots were prepared and submitted to the different inactivation procedures, while one aliquot was kept as an unprocessed control. Cytokine concentrations were measured according to the instructions provided with the assay kit (Bio-Techne, Minneapolis, MN, USA, Rat Luminex Discovery assay #LXSARM-05), and the data were subsequently analyzed and plotted in GraphPad Prism 10 (GraphPad Prism version 10.2.1, GraphPad Software, Boston, MA, USA).

### 2.11. LDH Assay

Lactate dehydrogenase (LDH) levels were quantified on culture media spiked with LDH of known concentrations using the CyQUANT LDH Cytotoxicity Assay Kit (Invitrogen, Waltham, MA, USA, #C20300). Briefly, 1 µL of the LDH-positive control solution (provided in the kit) was added to 1.5 mL of cell medium, and 100 µL of this mixture was aliquoted into individual Eppendorf tubes. Following inactivation with UVC light and 0.05% TX-100, 50 µL of the inactivated probes, as well as 50 µL of untreated control, was mixed with 50 µL of the reaction mixture. The samples were incubated for 30 min, after which, 50 µL of STOP solution was added to terminate the reaction. Absorbance was measured at 490 nm and 680 nm using the BioTek Cytation 5 plate reader as an index of LDH activity.

## 3. Results

### 3.1. BCA Assay

We first determined the protein concentration of our samples by using the bicinchoninic acid (BCA) assay. The protein content in the cell extract (cell medium with scrapped cells) was 3664 µg/mL, and after dilution with cell medium, it was 2780 µg/mL, close to the protein content of the cell medium alone (2686 µg/mL). For quantification, the homogenate was initially diluted to 1:5 to fall within the range of the standard curve, yielding a measured value of 1406 µg/mL, meaning that the protein content of the undiluted sample was 7030 µg/mL. Thus, the protein concentration in the raw homogenates was approximately 2.5 times higher than the one in the diluted cell medium (Appendix A).

### 3.2. Inactivation with UVC Light

UVC irradiation is commonly used for viral inactivation. We first evaluated the ability of UVC irradiation to inactivate TBEV in our different samples by varying the duration of exposure. In the initial experiment, the time required for complete inactivation was roughly determined. Cell supernatant with a viral concentration of 1.1 × 10^7^ PFU/mL was diluted to 1:10 with cell medium to have enough volume to test all conditions. Then, 100 ul of the samples, contained in a 1.5 mL PP clear tube, was exposed to UVC light for periods ranging from 15 s to 10 min. Titers, measured via TCID_50_, were strongly reduced after 15 s (~100 mJ/cm^2^), and the inactivation was fully achieved by 30 s (~300 mJ/cm^2^) or with a higher exposure time (Appendix A). The experiment was repeated by using exposure times between 15 and 90 s and replicated independently three times, using plaque assays for more precise viral quantification. After 15 s, the samples were non-infectious in two out of three biological replicates, with the remaining sample showing a viral load of 30 PFU/mL, corresponding to an estimated 4.5-log reduction. With 30 s or longer exposures, we could not detect any infectious viral particle (detection limit 10 PFU/mL) in any replicate, achieving a theoretical reduction in viral titers of at least 5-log (Figure 1A).

We next investigated whether 30 s of irradiation, shown to be sufficient to inactivate the virus in the culture supernatant, would also be effective in inactivating homogenates, which represent a more complex and protein-rich matrix. For this purpose, the tests were performed on different brain homogenates, given the limited amount of available material. After 30 s of UVC irradiation on an undiluted homogenate (initial concentration = 2.4 × 10^5^ PFU/mL), many plaques were still detected, indicating incomplete inactivation. After diluting several homogenates to 1:5, with final concentrations up to 7.0 × 10^6^ PFU/mL, complete inactivation was achieved within 30 s in all samples (Figure 1B), registering a theoretical 5.8-log viral reduction.

### 3.3. Mechanical Filtration

We next investigated the use of concentrator columns to filter viral particles from infected culture media, initially containing 1.1 × 10^7^ PFU/mL. Columns equipped with membranes of different pore sizes with cutoffs of 100 kDa and 300 kDa were tested. The 100 kDa column successfully retained all viral particles from the supernatant, while the 300 kDa column failed to retain TBEV (Appendix A). As the primary purpose of these columns was to concentrate the viral samples, we examined the viral titer of the concentrated sample above the filters with a cutoff value of 100 kDa. The measured viral concentration, performed via TCID_50_, was 1.4 × 10^8^ PFU/mL—representing an approximately 10-fold increase over the initial stock, therefore proving the retention of infectious viral particles by the 100 kDa membrane.

### 3.4. Chemical Inactivation

#### 3.4.1. Triton X-100 Inactivation

TX-100 is widely used as an organic detergent, particularly in protocols for protein extraction and cell membrane permeabilization. Notably, it is also effective at inactivating enveloped viruses by disrupting their lipid layers. Earlier studies determined that a 0.2% concentration of Triton achieves complete TBEV inactivation [6]. We can confirm the efficacy of TX-100 and show that even lower concentrations are able to inactivate TBEV in cell culture medium (TCID_50_ = 1.1 × 10^7^ PFU/mL). While 0.025% achieved full inactivation in only one of the three independent experiments, 0.05% Triton was sufficient to fully inactivate TBEV after an incubation of 20 min at room temperature in all three experiments (Figure 2).

Since a 0.05% concentration was sufficient to fully inactivate the virus in the cell supernatant, this concentration was also assessed on a different matrix composition, that is, homogenates. Notably, incubation for 20 min at room temperature with 0.05% Triton successfully inactivated two out of three different undiluted brain homogenates of TBEV-infected animals (Figure 3A). In the experiment not showing complete inactivation, a 5-log reduction (from 1.2 × 10^6^ PFU/mL to 20 PFU/mL) was nevertheless reached. In a new series of experiments, where the concentration of TX-100 was increased to 0.1%, it was possible to fully inactivate another set of three homogenates, with titers up to 2.4 × 10^8^ PFU/mL, therefore confirming an at least 7-log reduction, validating the procedure (Figure 3B).

#### 3.4.2. Tween-20 Inactivation

Tween-20 is a detergent commonly used in ELISA, Western blotting, and other immunoassays, so it was assumed that its presence in the sample should not interfere with the respective tests. This would, therefore, make it an ideal candidate for samples destined for such assays, provided it can inactivate TBEV. To address this, we tested a range of concentrations from 0.05% to 10% on the same viral stock as used for Triton inactivation experiments (1.1 × 10⁷ PFU/mL). Concentrations of Tween-20 below 10% were unable to fully inactivate the virus, although a dose-dependent decrease in the number of infectious viral particles was observed. A 10% concentration may achieve complete viral inactivation, but the results are subject to caution since the inactivated sample containing Tween may have been cytotoxic to the cells, as judged by an effect on the monolayer integrity (Figure 4).

### 3.5. Luminex Measurements

As described above, we have identified three effective methods for TBEV inactivation: UVC irradiation, column filtration, and treatment with 0.05/0.1% TX-100 (cell medium/brain homogenates). We first assessed whether these treatments preserved the compatibility of the samples for immunoassays. The different tests were performed on non-infectious material. For this purpose, cell medium was spiked with cytokines of known concentrations and subjected to each inactivation method. The level of several cytokines (IFN-γ, IL-6, TNF-α, IL-1β, and IL-10) was assessed by Luminex technology in the spiked medium before and after inactivation, and the rate of recovery after inactivation procedures was calculated.

For UVC irradiation, we chose a safe exposure time of 45 s, slightly longer than the validated duration of 30 s, allowing for variations in sample composition or viral titer that could affect inactivation time. For TX-100, we tested both 0.05% and 0.1% since both concentrations showed safe inactivation in cell medium.

Neither UVC exposure nor either concentration of TX-100 interfered with the immunoassay. Compared to the untreated control, the two methods resulted in a recovery rate between 76 and 112%. The variation was the highest for IL-10 (higher than control) and IFN-γ (lower than control). For the other cytokines, the values of the inactivated samples were close to the control value. In contrast, column filtration led to almost complete cytokine depletion, rendering it unsuitable for downstream analyses (Figure 5).

Based on these results, column filtration as an inactivation method was excluded from further testing. We next proceeded with experiments on homogenates, processed with a 45 s UVC exposure or treatment with either 0.05% or 0.1% TX-100. Homogenates from the brain of a non-infected animal, expected to contain low endogenous cytokine levels, were spiked with exogenous cytokines, similar to what was performed for cell culture supernatant. We applied the three inactivation methods to these samples. The concentrations of IL-6, TNF-α, and IL-1β were comparable to the non-processed control (ranging from 80% to 103%), while IL-10 (recovery: 99% to 126%) and IFN-γ (recovery: 50% to 108%) showed greater variability, consistent with previous results using culture medium [Figure 6]. Of note, the higher TX-100 concentration (0.1%) had a greater impact on the sample recovery than 0.05% TX-100 or UVC irradiation, with UVC irradiation preserving cytokine concentrations more effectively. These results, therefore, confirm the suitability of these approaches for downstream immuno-based analyses.

### 3.6. LDH Assay

To assess whether the inactivation procedures affected enzymatic activity, we selected a lactate dehydrogenase (LDH) assay, often used to test cytotoxicity in the culture supernatants of infected cells. In this assay, we spiked the cell medium with recombinant LDH and subjected the samples to either a 45 s UVC exposure or treatment with 0.05% TX-100, both showing effective inactivation in culture supernatants. TX-100 inactivation resulted in a recovery rate of 119%, while UVC for 45 s resulted in a slightly lower, but less variable, recovery rate of 92% (Figure 7). Both inactivation methods enabled us to measure LDH levels in the samples, confirming their suitability for downstream enzymatic activity assay.

## 4. Discussion

This study is primarily aimed at researchers involved in cell culture or *in vivo* experimental models, particularly those investigating host responses to TBEV. These types of studies often require the processing and analyses of infectious material outside of BSL-3 facilities, particularly for downstream applications such as flow cytometry, cytokine profiling, or proteomics, which cannot always be conducted within high-containment laboratories. In particular, the antigenic and enzymatic properties of the samples, which are essential for immune-based or enzymatic analyses, should remain unaltered. For this specific purpose, the inactivation methods were tested on relatively small volumes (100 µL) that are usually sufficient for downstream analyses. We did not focus on diagnostic workflows aimed at detecting TBEV or TBEV-specific antibodies, as the associated biosafety requirements may differ when the virus is not intentionally propagated in diagnostic samples.

### 4.1. UVC, Triton-X100, and Column Filtration Successfully Inactivate TBEV-Infected Probes with Different Protein Contents

UVC irradiation at a wavelength of 254 nm has been shown to effectively inactivate TBEV. In samples with a relatively simple matrix and low protein content (3.7 mg/mL), such as cell culture supernatants with lysed cells, a brief exposure of 15 s (100 mJ/cm^2^) resulted in a 4.5-log reduction in viral concentration. Extending the exposure to 30 s further reduced the viral titer below the detection limit of the plaque assays (10 PFU), resulting in at least a 5-log reduction. Similar exposure conditions have been shown to be effective for inactivating Chikungunya virus, an enveloped alphavirus [23]. Based on these findings, we expect that other cell culture supernatants, typically harvested without significant cell lysis and, therefore, containing even lower protein concentrations, would be even more efficiently inactivated by UVC irradiation.

On raw tissue homogenates (7.0 mg/mL proteins) constituting a more complex matrix, we observed that UVC’s effectiveness was greatly reduced. A 30 s exposure was not able to fully inactivate TBEV. However, following a five-fold dilution of the raw homogenate (reducing protein content to 1.4 mg/mL), the 30 s UV exposure reduced the viral titer by almost 5.8 log, bringing it below our detection limit (10 PFU). This was expected, as proteins and other matrix components can absorb light at 254 nm, limiting UVC penetration and, consequently, its inactivation efficiency [24,25]. Indeed, even common cell culture media, such as DMEM and fetal bovine serum, have been reported to slightly absorb UVC radiation [13,25].

Another important factor affecting inactivation efficiency is the distance between the light source and the sample. In our study, we used a UVC crosslinker, which maintains a short distance between the light source and the samples. Other studies have used UV lamps in biosafety cabinets, sometimes without mentioning distance or measuring the effective irradiation energy. The increased distance or the use of UV light sources with reduced light intensity due to extended use may impact inactivation efficiency [13,16].

Further, our inactivation experiments were performed using sample volumes of 100 µL, a format compatible with Luminex and enzymatic assays conducted in 96-well plates. Depending on needs, this method should, therefore, be adapted and validated for larger volumes since this most likely will alter the requirements for complete viral inactivation. Finally, the composition of the vessel containing the viral suspension is important, particularly when the vessel is sealed. Polypropylene, for instance, is known to partially absorb UVC wavelengths, meaning that irradiation time should be adjusted accordingly. An alternative approach would be to either irradiate samples in open vessels, such as multi-well plates, or to use vessels made from a different UV-transparent material [26].

The chemical inactivation of TBEV stocks has previously been demonstrated using Tween-20 and Triton X-100, both of which can reduce RNA copy numbers up to 5 log at concentrations of 0.2% and 5%, respectively, although the influence of the cytotoxicity of the compounds cannot be excluded [13]. In our experiments, we tested various concentrations of these reagents and found that Triton X-100 effectively inactivated TBEV in cell culture supernatants at concentrations ranging between 0.05% and 0.1%. However, for the homogenates, a minimum concentration of 0.1% was required for complete inactivation. This was probably due to the higher complexity of the matrix components.

In contrast, we were unable to determine the minimum effective concentration of Tween-20 needed for TBEV inactivation. A 5% concentration was insufficient, while 10% exhibited cytotoxic effects on cells in our experimental setup. Interestingly, previous studies have reported that a washing buffer containing 0.05% Tween-20 successfully inactivated WNV within 30 min at a temperature of 37 °C [19]. It would be, therefore, interesting to test whether increasing the incubation time and/or temperature would more effectively inactivate the samples without impacting the downstream analyses. Further, recent concerns about the undesired effect of the degradation products of TX-100, which are known endocrine disruptors toxic to aquatic organisms, have been raised, so replacement methods should also be evaluated in the future. In that context, non-ionic detergents, such as Simulsol SL 11W, for example [27], could represent a valuable alternative that could be tested.

As a third potential method, we also tested mechanical filtration using 100 and 300 kDa membranes. Only membranes with a cutoff of 100 kDa were able to retain infectious viral particles from the flow-through while concentrating them in the non-filtered fraction of the medium. This was also confirmed by another study, where TBEV was removed from the supernatant of TBEV-infected primary human brain cortical astrocytes (HBCAs) under similar conditions [14].

While the number of biological replicates per condition was limited, consistent results were obtained across independent experiments. These findings provide a robust foundation for the proposed inactivation methods, though future studies using larger sample sets and broader conditions could further support their generalizability.

### 4.2. UVC and Triton-X100 Inactivation Preserve Protein Integrity for Downstream Analyses

Having identified three effective inactivation methods, 45 s of UVC exposure, 0.05%/0.1% TX-100, and 100 kDa column filtration, we evaluated their compatibility for use in immunoassays. To ensure compatibility, the epitopes of the target protein must be preserved. It is known that UVC light may have a detrimental effect on proteins [28], while Triton-X100, as a non-ionic detergent, has shown generally good protein preservation, although denaturation may occur.

Using cell supernatant spiked with the cytokines of known concentrations, we demonstrated that UVC irradiation and TX-100 processing largely preserved the detection of all five cytokines, with recovery rates close to the control values. We showed that the recovery rate after UVC exposure was in a range of 96-100%, while the one in TX-100-treated samples was more variable, ranging between 76 and 112%, with the highest variability displayed by IFN-γ. In contrast, column filtration resulted in the near-complete loss of proteins with recovery rates of 3-13%, presumably due to depletion. This lower recovery rate upon ultrafiltration was also demonstrated in a study about the impact of decontamination methods on the cytokine profiles of samples of COVID-19 patients [26].

We also tested UV- and TX-100-processed brain homogenates diluted to 1:5 with saline buffer that were spiked with cytokines. In this setup, IFN-γ exhibited the largest variation compared to the control, while the other four cytokines (IL-6, IL-1β, TNF-α, and IL-10) displayed values closer to the control. This was in line with what was observed in the cell medium measurements. Also, in this case, the samples processed with UVC showed values closer to the control (99–108%), while TX-100 showed greater variability (50–126%), depending on the concentration of the detergent. These results are in line with the observations obtained with the inactivation of SARS-CoV-19 probes. Particularly, it was shown that inactivation via UV light or TX-100 showed good recovery for the cytokines after viral inactivation [26,29].

These findings emphasize the importance of carefully interpreting cytokine measurements after the processing of samples for inactivation, particularly in terms of absolute or relative quantification. For cytokines like IL-6, IL-10, IL-1β, and TNF-α, we observed minor variations compared to unprocessed samples, so that absolute values are still relevant. However, the levels of cytokines such as IFN-γ, which showed greater variability after inactivation, should be interpreted as relative values and compared to proper experimental controls. Therefore, we recommend systematically pre-testing the chosen inactivation method for its impact on target analytes before conducting large-scale experiments.

### 4.3. UVC and Triton-X100 Inactivation Do Not Interfere with Enzymatic Activity

We also evaluated whether UVC irradiation and TX-100 treatments affected enzymatic activity during LDH assay, a method commonly used to estimate cytotoxicity in cell culture experiments. Here, we observed a difference of up to 20% compared to the unprocessed control sample. The presence of TX-100 in the sample appeared to enhance reaction speed, which was also in line with other studies showing an increase in enzymatic activity [30], whereas the UV treatment produced lower values. These observations reinforce the importance of interpreting these measurements as relative values in comparison to proper experimental controls, as results in terms of absolute values may be influenced by the inactivation method.

### 4.4. Comparative Applicability and Limitations of Inactivation Methods

To support researchers in selecting the most suitable method for their specific applications, we briefly outline the practical applicability and key limitations of each inactivation strategy. UVC irradiation is reagent-free and rapid and generally preserves protein integrity, making it well suited for immunoassays and enzymatic tests. However, it requires access to a calibrated UVC crosslinker or an equivalent device in order to precisely generate the required irradiation doses. Further, its effectiveness is influenced by sample composition, volume, and the material of the container. Triton X-100, in contrast, is independent of sample volume and less affected by sample complexity. Its main limitations lie in the compound’s toxicity and in the observed variability in some protein measurements. However, this limitation can be resolved when results are interpreted in a relative manner through the comparison of the inactivated samples of the different experimental conditions under the same conditions.

## 5. Conclusions

This study demonstrates that TBEV can be effectively inactivated using UVC irradiation; Triton X-100 treatment; and, to a lesser extent, mechanical filtration. Both UVC exposure (30 s) and Triton X-100 at concentrations of 0.05–0.1% achieved complete viral inactivation and preserved compatibility with downstream assays such as cytokine profiling and enzymatic activity measurement. Among the methods tested, UVC irradiation proved to be the most conservative, with recovery values closely matching those of the untreated controls (±8%), although its efficacy was influenced by sample composition and light-absorbing matrix components. Triton X-100 provided robust inactivation across sample types and was less affected by matrix composition but exhibited greater variability in downstream protein measurements (up to ±50%). In contrast, filtration through 100 kDa membranes removed viral particles effectively but caused significant protein depletion, limiting its utility for proteomic or immunological analyses.

Our study did not investigate the long-term stability of inactivated samples. This decision reflects the practical workflow of our experimental design. Specifically, we focused on low-volume samples compatible with 96-well plate-based assays, where inactivation typically occurs immediately prior to analysis, thereby reducing the need for extended storage. Nevertheless, we recognize that long-term stability may be relevant in certain research and diagnostic contexts. Previous studies have evaluated the stability of cytokines and other analytes, with results showing varying degrees of degradation depending on the specific target, storage conditions, and storage duration [31]. Future work could address how different inactivation methods affect sample stability during longer-term storage.

Together, these findings support UVC irradiation and Triton X-100 treatment as simple, effective, and broadly applicable methods for the safe inactivation of TBEV in samples destined for protein-based assays in research settings. These methods may also be adaptable for use with other enveloped viruses, provided they are validated under appropriate biosafety conditions. We recommend that any selected inactivation protocol be empirically tested in the context of the specific experimental setup and target analytes prior to implementation.

## Figures and Tables

**Figure 1 viruses-17-00810-f001:**
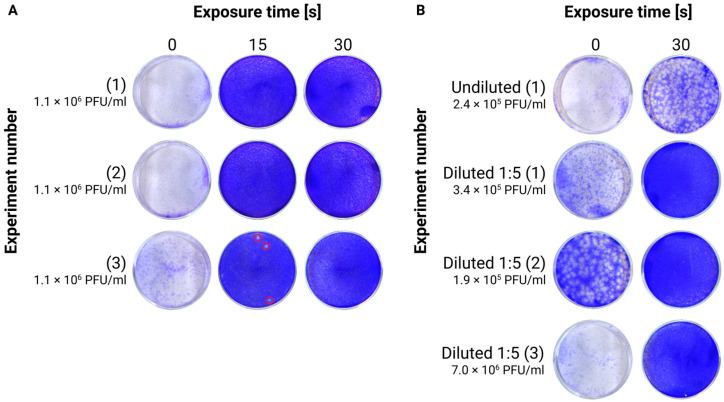
Testing of the efficiency of UVC irradiation for TBEV inactivation by plaque assays. (**A**) The infected cell supernatant, with an initial concentration of 1.1 × 10^6^ PFU, was exposed to UVC light for different durations, spanning 15 to 90 s. The irradiated cell supernatant was then used as inoculum for the plaque assay. The experiment was repeated 3 times and reported only 15 and 30 s conditions. Red circles highlight the presence of plaques; (**B**) different homogenates (undiluted and diluted to 1:5), with different initial concentrations, were exposed to UVC light for 30 s. The irradiated homogenates were then used as inoculum for the plaque assay.

**Figure 2 viruses-17-00810-f002:**
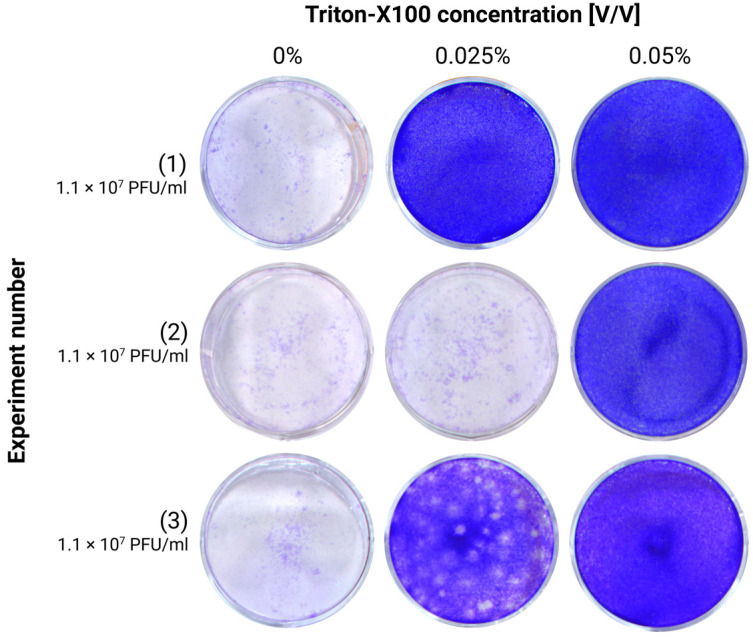
Testing of the efficiency of TX-100 treatment for TBEV inactivation in cell supernatants by plaque assays. The infected cell supernatant, with an initial concentration of 1.1 × 10^7^ PFU, was treated with Triton-X100 to a final concentration of 0.025% and 0.05%. The treated cell supernatant was then used as inoculum for the plaque assay. The experiment was repeated 3 times.

**Figure 3 viruses-17-00810-f003:**
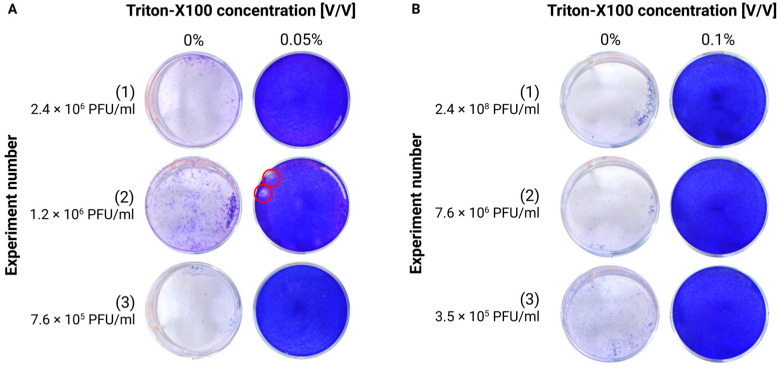
Testing of the efficiency of Triton-X100 treatment for TBEV inactivation in brain homogenates by plaque assays. (**A**) Brain homogenates were treated with Triton-X100 to a final concentration of 0.05%. The treated homogenates were then used as inoculum for the plaque assay. The experiment was repeated 3 times with different starting homogenates. Red circles highlight the presence of plaques; (**B**) brain homogenates were treated with Triton-X100 to a final concentration of 0.1%. The treated homogenates were then used as inoculum for the plaque assay. The experiment was repeated 3 times with different starting homogenates.

**Figure 4 viruses-17-00810-f004:**
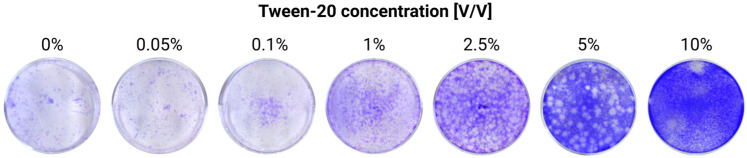
Testing of the efficiency of Tween-20 treatment for TBEV inactivation in cell supernatants by plaque assays. Infected cell supernatants, with an initial concentration of 1.1 × 10^7^ PFU/mL, were treated with Tween-20 to final concentrations of 0.05, 0.1, 1, 2.5, 5, and 10%. The treated cell supernatants were then used as inoculum for the plaque assay. The experiment was repeated once.

**Figure 5 viruses-17-00810-f005:**
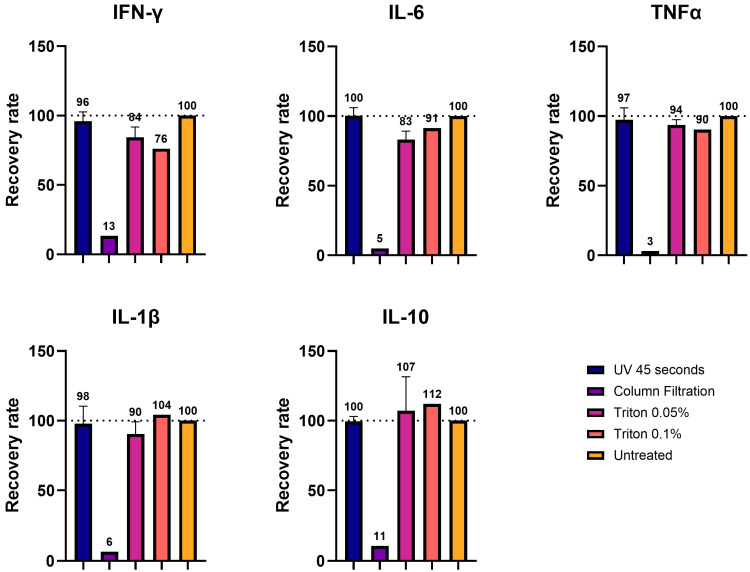
Determination of the impact of inactivation methods applied to cell supernatants on Luminex immunoassays. Cell supernatants were inactivated with different procedures: UV 45 s, column filtration, 0.05% Triton-X100, and 0.1% Triton-X100. Five different cytokines (IFN-γ, IL-6, TNFα, IL-1β, and IL-10) were then measured using the Luminex, and their values were expressed in recovery rates (%) as a function of the control (100%).

**Figure 6 viruses-17-00810-f006:**
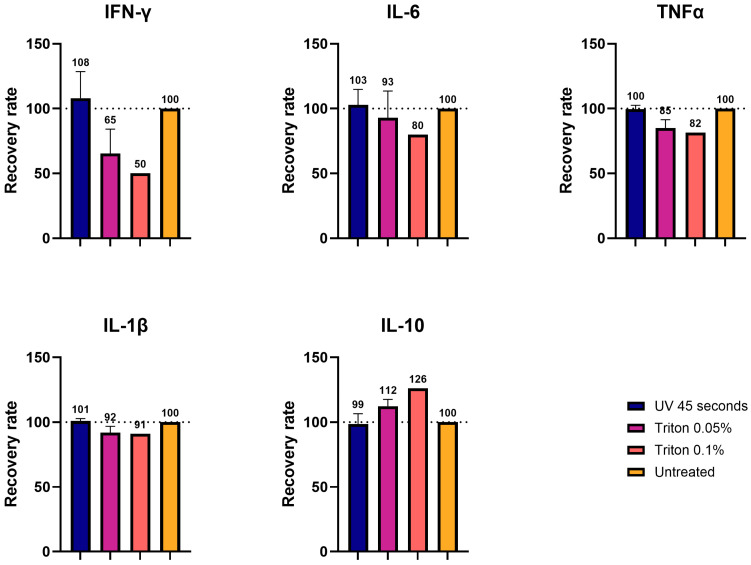
Determination of the impact of inactivation methods applied to brain homogenates on Luminex immunoassays. Brain homogenates were inactivated with different procedures: UV 45 s, 0.05% Triton, and 0.1% Triton. Five different cytokines (IFN-γ, IL-6, TNFα, IL-1β, and IL-10) were then measured using the Luminex, and their values were expressed in recovery rates (%) as a function of the control (100%).

**Figure 7 viruses-17-00810-f007:**
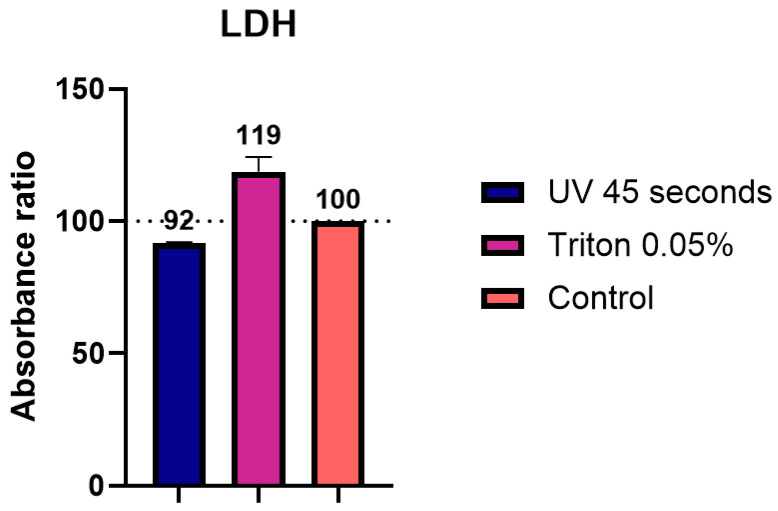
Determination of the impact of inactivation methods applied to cell supernatants on LDH assays. Cell supernatants were inactivated with different procedures: UV 45 s and 0.05% Triton. The absorbance ratio was then measured with a plate reader and expressed in % as a function of the control value (100%).

## Data Availability

The raw data supporting the conclusions of this article will be made available by the authors upon reasonable request.

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
