# Peer review of "Validated Methods for Inactivation of Tick-Borne Encephalitis Virus Compatible with Immune-Based and Enzymatic Downstream Analyses"

_viruses, 2025, doi:10.3390/v17060810_

Round 1
Reviewer 1 Report
Comments and Suggestions for Authors
The Tick-Borne Encephalitis Virus (TBEV) poses a serious threat to public health in the Eurasian region, with an increasing number of cases year by year. Researching effective virus inactivation methods helps improve laboratory safety, reduce the risk of laboratory infections, and is of significant public health importance. This manuscript systematically evaluates various inactivation methods, including ultraviolet (UV) irradiation, chemical reagents (such as Triton X-100), and mechanical filtration. It holds certain reference value. But there are many flaws.
1.The sample size used in the study is relatively small, which may be insufficient to comprehensively represent the actual conditions under various experimental scenarios, potentially affecting the generalizability of the results.
2.Although the study evaluated multiple inactivation methods, the specific applicability and limitations of each method were not discussed in detail, which may affect the reader's judgment in choosing the appropriate method.
3.Lack of long-term stability data: The study mainly focused on the immediate effects of inactivation methods, without evaluating the long-term stability of samples after inactivation, which may affect the reliability of the methods in practical applications.
4. Enhance the quality and aesthetic appeal of the figures, and replace the summary figure.
5.The writing format is not standardized, such as in the discussion section 4.1.
6.Add the latest relevant references.
Comments on the Quality of English Language/
Reviewer 2 Report
Comments and Suggestions for Authors
This study investigates different inactivation methods for the tick-borne encephalitis virus suitable for further analysis outside a biosafety level 3 laboratory. The experiments are well described, and the results are clearly presented. Unfortunately, I have some difficulties understanding the goal or the conclusions throughout the reading of the manuscript.
You show that the inactivation methods do not seem to have any influence on the analysis of various laboratory parameters (when spiking the samples), but I struggle with the following points:
- The course of an infection with TBEV is usually biphasic, with viremia and unspecific symptoms in the first phase, followed by severe disease in the second phase. The blood draw is usually made in the second phase, where no more viruses are detectable, and diagnosis is mainly based on antibody determination.
- You also write that you want to preserve the antigenic and enzymatic properties for immune-based or enzymatic analysis. This might be true for analysing some parameters but not for TBEV. To show that you can still enzymatically detect TBEV in your samples or use your samples as antigens, you need to perform further analysis (e.g., ELISA).
Overall, inactivating TBEV is an important task in viral research, but I suggest specifying the objectives, conclusions, and discussion regarding the points mentioned above.
Round 2
Reviewer 2 Report
Comments and Suggestions for Authors
All my suggestions from the initial review report have been addressed by the authors.